# Quantitative Aspect of *Bacillus subtilis* σ^B^ Regulatory Network on a Proteome Level—A Computational Simulation

**DOI:** 10.3390/biology13080614

**Published:** 2024-08-13

**Authors:** Jiri Vohradsky

**Affiliations:** Laboratory of Bioinformatics, Institute of Microbiology, Czech Academy of Sciences, Vídeňská 1083, 142 20 Prague, Czech Republic; vohr@biomed.cas.cz

**Keywords:** *Bacillus subtilis*, sigma B, computational modeling, protein regulatory networks

## Abstract

**Simple Summary:**

*Bacillus subtilis* is a model organism used to study molecular processes in Gram-positive bacteria. Sigma factor B (σ^B^, SigB) is a major regulator of gene expression in response to various stresses. SigB itself is controlled by a network involving several other factors. In this paper, I focused on computational modeling of their interactions and analyzed how these interactions influence the production of SigB and other components of the network. Understanding the detailed functioning of such a network and the concept of its analysis helps to comprehend molecular processes occurring in the cell.

**Abstract:**

*Bacillus subtilis* is a model organism used to study molecular processes in Gram-positive bacteria. Sigma factor B, which associates with RNA polymerase, is one of the transcriptional regulators involved in the cell’s response to environmental stress. Experiments have proven that the amounts of free σ^B^ (SigB) are controlled by a system of anti- (RsbW) and anti-anti-sigma (RsbV) factors expressed from the same operon as SigB. Moreover, the phosphorylation state of RsbV is controlled by phosphatases RsbP and RsbU, which directly dephosphorylate RsbV. A set of chemical equations describing the network controlling the levels of free SigB was converted to a set of differential equations quantifying the dynamics of the network. The solution of these equations allowed the simulation of the kinetic behavior of the network and its components under real conditions reflected in the time series of protein expression. In this study, the time series of protein expression measured by mass spectrometry were utilized to investigate the role of phosphatases RsbU/RsbP in transmitting the environmental signal. Additionally, the influence of kinetic constants and the amounts of other network components on the functioning of the network was investigated. A comparison with the same simulation performed using a transcriptomic dataset showed that while the time series between the proteomic and transcriptomic datasets are not correlated, the results are the same. This indicates that when modeling is performed within one dataset, it does not matter whether the data come from the mRNA or protein level. In summary, the computational results based on experimental data provide a quantitative insight into the functioning of the SigB-dependent circuit and offer a template for the quantitative study of similar systems.

## 1. Introduction

The transcription factor in *Bacillus subtilis* has been extensively studied for its pivotal role in orchestrating the general stress response [1]. This alternative sigma factor activates the transcription of a diverse set of genes, enabling the bacterium to adapt to a wide range of environmental stresses, including heat, osmotic stress, or nutrient deprivation [2,3,4,5].

Early studies on SigB primarily focused on its regulation and activation mechanisms. Research by Hecker et al. [6] highlighted the role of SigB in stress-induced gene expression and identified key components of its regulatory network. SigB itself is regulated by a mechanism involving anti-sigma and anti-anti-sigma factors and additionally by phosphatases that act upstream of the “anti” factors. Under regular (stress-free) growth conditions, SigB is inactivated by binding of anti-σ factor RsbW. When the cell detects stress, the phosphatases RsbP and/or RsbU dephosphorylate RsbV, which subsequently binds to RsbW and SigB is released from RsbW. Free SigB is able to form holoenzyme with RNA polymerase, leading to transcription initiation of SigB-dependent genes, including its own operon [7,8,9,10]. These studies established the foundation for understanding the hierarchical and dynamic nature of SigB regulation.

In recent years, advancements in transcriptomic and computational biology have facilitated more comprehensive analyses of the SigB regulon [3,4,11,12]. Complementary to experimental techniques, computational simulations have emerged as powerful tools to model and predict the behavior of regulatory networks. For instance, Narula et al. [13] employed mathematical modeling to simulate SigB activation dynamics, providing a quantitative framework to study its regulatory properties. Kinetic analysis of the SigB regulon was presented in our previous study [14].

Despite significant progress, there remains a gap in fully understanding the SigB impact on the global protein level. This paper builds on my previous analysis and simulation of the SigB regulatory network performed using transcriptomic data [15]. From this paper, I adopted chemical and mathematical equations and used them in combination with original proteomic time series (PRIDE accession number PXD048690 [16]) in order to obtain the results related to protein expression.

This study aims to build on these foundations by providing a detailed quantitative analysis of the SigB regulatory network at the proteome level through computational simulations. By leveraging advanced computational tools and comprehensive proteomic datasets, I seek to elucidate the dynamic interactions and regulatory mechanisms controlling SigB, thereby enhancing our understanding of bacterial stress response strategies.

## 2. Materials and Methods

### Data Acquisition

Previously published data were used (PRIDE accession number PXD048690). The basic analysis of the dataset is given in Pospisil et al. [16]. Briefly, the spore fraction of the bacterial culture at time zero was determined using the heat shock method. *B. subtilis* culture grew in defined liquid medium. Samples for mass spectrometry were collected at ten-minute intervals (0, 10, 20, 30, 40, 50, 60, 70, 80, 90, 100, 110, 120, 130 min) and experiments were run in three biological replicates. A total of 2,191 proteins were identified and quality control passed 2063 unique proteins. The values at each time point for each individual protein were averaged to form time series of protein expression. The resulting time series for individual proteins were smoothed and subsampled to one-minute time interval using B-spline, as coded by Jonas Lundgren [17], and the time series for SigB, RsbW, RsbV, RsbU, and RsbP were used in the presented simulations. Their time series are shown in Appendix A.

## 3. Results

The SigB regulatory network consists of five principal factors: SigB, anti-sigma factor RsbW, anti-anti-sigma factor RsbV, and two phosphatases, RsbP and RsbU. The regulation starts at anti-anti-sigma factor RsbV, which can be dephosphorylated by two phosphatases, RsbP and RsbU. RsbW readily dimerizes to RsbW_2_, which is bound by dephosphorylated RsbV to release SigB from its complex with RsbW. RsbW also has a protein kinase activity that phosphorylates and inactivates RsbV (the complete scheme comprises several more chemical reactions and is given by Equations (ch1)–(ch8) and shown in Figure 1).

### 3.1. The Model

The SigB network is formally described by a set of biochemical Equations (ch1)–(ch8) [13] and the model is formed by differential Equations (a1)–(a10) [15] derived from the chemical equations capturing dynamics of the system. The following equations describe interactions among the molecules of the network (the anti-sigma factor RsbW readily dimerizes and acts in the biochemical reactions as a dimer [18]).

1.Binding of anti-sigma RsbW_2_ and anti-anti-sigma RsbV
(ch1)$$RsbW_{2}+RsbV\underset{k_{1-}}{\overset{k_{1+}}{⇄}}RsbW_{2}RsbV$$
(ch2)$$RsbW_{2}Rsbv+RsbV\underset{k_{2-}}{\overset{k_{2+}}{⇄}}RsbW_{2}RsbV_{2}$$2.Phosphorylation of anti-anti-sigma factor by anti-sigma factor
(ch3)$$RsbW_{2}+RsbV+P\overset{k_{3+}}{\to }RsbW_{2}+RsbV-P$$
(ch4)$$RsbW_{2}RsbV_{2}+P\overset{k_{4+}}{\to }RsbW_{2}RsbV+RsbV-P$$3.Binding of sigma factor to anti-sigma factor
(ch5)$$RsbW_{2}+SigB\underset{k_{5}-}{\overset{k_{5+}}{⇄}}RsbW_{2}SigB$$4.Displacement of SigB by RsbV in complex with RsbW_2_
(ch6)$$RsbW_{2}SigB+Rsbv\underset{k_{6-}}{\overset{k_{6+}}{⇄}}RsbW_{2}Rsbv+SigB$$5.Dephosphorylation of anti-anti-sigma factor by RsbU RsbP
(ch7)$$RsbVP+RsbU\overset{k_{7+}}{\to }RsbV+RsbU+P$$
(ch8)$$RsbVP+RsbP\overset{k_{7+}}{\to }RsbV+RsbP+P$$

The measured values of SigB, RsbV, and RsbW were considered to be composed of the free forms together with the forms bound in the complexes (Equations (a1)–(a3)), as shown by the chemical equations above. The measured expression profile includes both synthesis and degradation that, therefore, do not have to be considered separately. In the next equation, the following abbreviated notation will be used: V = RsbV, W = RsbW, S = measured total amount of SigB, AS = measured total amount of RsbW, AAS = measured total amount of RsbV. Index *f* = free form of the factor.

The measured amounts of SigB, AS, and AAS comprised its free form and the form bound in the complexes.
(a1)$$\left[S]={[SigB}_{f}]+{[W}_{2}SigB\right]$$
(a2)$$\left[AS]=2{[W}_{2f}]+2{[W}_{2}V]+2{[W}_{2}V_{2}]+2{[W}_{2}SigB\right]$$
(a3)$$\left[AAS]={[V}_{f}]+{[W}_{2}V]+2\left[W_{2}V_{2}\right]+[VP\right]$$

By differentiating Equations (a1)–(a3) and the kinetic equations of (ch1)–(ch8), we obtain
(a4)$$\frac{{d[SigB}_{f}]}{dt}=\frac{d[S]}{dt}-\frac{d[W_{2}SigB]}{dt}$$
(a5)$$\frac{{d[W}_{2f}]}{dt}=\frac{1}{2}\frac{d[AS]}{dt}-\frac{d{[W}_{2}V]}{dt}-\frac{d{[W}_{2}V_{2}]}{dt}-\frac{d{[W}_{2}SigB]}{dt}$$
(a6)$$\frac{{d[V}_{f}]}{dt}=\frac{d[AAS]}{dt}-\frac{d[W_{2}V]}{dt}-2\frac{d[W_{2}V_{2}]}{dt}-\frac{d[VP]}{dt}$$
(a7)$$\begin{array}{ll} \frac{d[W_{2}SigB]}{dt}= & k_{5+}{[W}_{2}][{SigB}_{f}]-k_{5-}\left[W_{2}SigB\right]\\ & \;\;-k_{6+}\left[W_{2}SigB\right]{[V}_{f}]+k_{6-}{[W}_{2}V][S{igB}_{f}] \end{array}$$
(a8)$$\begin{array}{ll} \frac{d{[W}_{2}V]}{dt}= & k_{2-}{[W}_{2}V_{2}]-k_{2+}\left[W_{2}V\right][V_{f}]+k_{4+}{[W}_{2}V_{2}]\\ & \;\;+k_{1+}[W_{2f}]{[V}_{f}]-k_{1-}{[W}_{2}V]\\ & \;\;+k_{6+}[W_{2}SigB][V_{f}]-k_{6-}[W_{2}V][{SigB}_{f}] \end{array}$$
(a9)$$\frac{{d[W}_{2}V_{2}]}{dt}=k_{2+}\left[W_{2}V\right][V_{f}]-k_{2-}[W_{2}V_{2}]-k_{4+}[W_{2}V_{2}]$$
(a10)$$\begin{array}{ll} \frac{d[VP]}{dt}= & k_{3+}{[W}_{2f}][V_{f}]+k_{4+}[W_{2}V_{2}]-k_{7+}[VP][RsbP]\\ & \;\;-k_{7+}[VP][RsbU] \end{array}$$

The constants k_1_–k_7_ are given in Table 1.

Expression profiles and their derivatives of SigB, RsbW, RsbV (S, AS, and AAS), RsbP, and RsbU are readily available and their expression profiles are shown in Figure 2 and are available in Appendix A. Equations (a4)–(a10) were solved numerically in Matlab.

In order to determine the reaction of the network to varying amounts of the components of the network and the values of the kinetic constants, the profiles in Figure 2 were used in the simulations shown below.

### 3.2. Influence of RsbU

In the following paragraphs, I tested the influence of the factors controlling the amount of free SigB. As Figure 1 and the references mentioned in the Introduction show, the free SigB amount is determined by the phosphatases RsbU and RsbP, which dephosphorylate the anti-anti-sigma factor RsbV. RsbV then binds to the anti-sigma factor, freeing SigB. Therefore, at first, I tested the influence of changes in the amounts of RsbU. As mentioned above, the amount of RsbP was so low that it did not contribute to the regulation and was not considered in the simulations. The results of the simulation for the changing amounts of RsbU are shown in Figure 3.

Figure 3 shows that with the given amount of RsbU, almost all RsbV was dephosphorylated. With the decreasing amount of RsbU, the amount of phosphorylated RsbV increased and, consequently, the anti-anti-sigma factor RsbW_2_ was released from its complexes with RsbV, and RsbW_2_ could bind more of the sigma factor SigB. As a result, the amount of free SigB decreased. It also can be seen that while RsbU amounts ranged from their maximal value down to almost zero, the resulting amounts of SigB were influenced in a much lower range. This was caused by additional mutual reactions of anti- and anti-anti- factors which did not allow a full block of SigB by RsbW_2_, which is partially bound in other complexes or exists in a free form given by the corresponding chemical equations.

### 3.3. Influence of Anti-Anti-Sigma Factor RsbV

There are two main players influencing free SigB amount: anti-anti-sigma factor RsbV and anti-sigma factor RsbW_2_. The influence of RsbV can be studied by altering its amount and/or altering the rate constant of its dephosphorylation k_7+_. Figure 4 shows the influence of decreasing the amount of RsbV by 2-, 3-, and 5-fold.

Figure 4 shows that a decreasing amount of AAS increased the amount of free anti-sigma factor RsbW_2_ and the complex of RsbW_2_ with SigB. Consequently, the amount of free SigB decreased. The amounts of free RsbV and RsbW_2_ remained relatively high, limiting the range of the free SigB.

Another way of influencing the amount of active anti-anti-sigma factor is to alter the rate constant of RsbVP dephosphorylation by RsbU (Equation (ch7)). The results are shown in Figure 5, where the rate constant k_7+_ was divided by 5, 10, and 100, shifting the reaction in Equation (ch7) against phosphorylation of RsbV.

With the decreasing rate constant k_7+_, the amount of phosphorylated RsbV increased, the amount of its free form decreased, and the amount of free RsbW_2_ increased. The increase of RsbW_2_ means that more of SigB can be bound in the complex with RsbW_2_ and, as a result, the amount of free SigB decreased.

Altering the amounts of the free anti-anti-sigma factor, by changing its amount or changing the rate of its dephosphorylation, changed the amount of free SigB, but only in certain limits, which are given by the reactions controlling synthesis of the other complexes.

### 3.4. Influence of Anti-Sigma Factor RsbW_2_

Anti-sigma factor RsbW_2_ is considered, together with RsbU, as the main player in controlling the amounts of free SigB. To investigate the reaction of the system on changes in the amount of RsbW_2_, I made the simulations by multiplying AS by 2 and 5 and dividing it by 2. The results are shown in Figure 6.

The decrease/increase in the amount of the anti-sigma factor was reflected in free SigB, as expected, by the increase/decrease of the amount of free SigB. The increase of AS led to an increase of the amount of the complex SigBRsbW_2_ and consequent decrease of the amount of free SigB. Simulation resulted in a large effect; increasing the amount of anti-sigma factor can result in almost complete attenuation of SigB.

### 3.5. Influence of Rate Constants of the Reaction of SigB with Its Anti-Sigma Factor

The last reaction directly influencing the amount of free SigB is its reaction with anti-sigma factor RsbW_2_, given by Equation (ch5), with the rate constants k_5+_ and k_5-_ which allow the shift of the direction of the reaction towards the formation of the complex SigBRsbW_2_. Figure 7 shows how the profiles of the components of the system changed when the reaction was shifted by changing the rate constants by multiplication of k_5+_ by 10 and 100, while k_5-_ was divided by the same numbers.

Figure 7 shows that the shift of the reaction of SigB with RsbW_2_ resulted in the formation of more of the complex SigBRsbW_2_ and a decrease of the amount of free SigB, as expected, but in a range lower than expected.

### 3.6. Comparison of Results Obtained from Proteomic and Transcriptomic Datasets

The same analysis performed here was also undertaken with the dataset obtained from a transcriptomic experiment [15]. Figure 8 shows a comparison of the results of simulation for changes in the amounts of phosphatases RsbU, here, and RsbU and RsbP, for the transcriptomic experiment (the colors in the figure were organized so that they were the same for the same trend, red—highest, green—middle, blue/cyan—lowest). Figure 8 shows similar trends—the increasing/decreasing amounts of phosphatases are reflected in the same way in both experiments. The line color order in the graphs of both panels exhibit the same trends. Also, the effect on the final amount of free SigB is only limited in comparison with the scale of the changes in the phosphatases. It shows that the crucial aspect of the network is the mutual relationship among the molecules and their expression time series within the system, rather than whether the data come from proteomic or transcriptomic sources.

## 4. Discussion

The model not only confirms the experimentally observed behavior of the system, which is a trivial requirement for the model to be working, but also allows seeing quantitatively how the amounts of each of the components of the network develop over time and during the simulations altering the parameters of the system. The simulations given above show that the most important parameters of the system were the activity of the phosphatases, here RsbU, and the activity of the anti-sigma factor RsbW. If one wants to control SigB regulation, the most effective way is to overproduce the anti-sigma factor, which can attenuate the production of SigB entirely. The influence of other components of the system is moderate. Another phenomenon was observed during the simulation and is related to the initial values used during the solution of the differential equations. While the initial values of the proteins SigB, RsbW_2_, and RsbU were given by the value at time zero, the value of RsbVP must have been set to the AAS at time point zero. This corresponds to the situation at the beginning when all of the RsbV was phosphorylated. If the initial value for RsbVP is set to lower values, the solution of the differential equations creates artifacts. This suggests that, in order for the system to work in reality, the RsbV should also be fully (or at least highly) phosphorylated. This should be proven experimentally.

The same analysis presented here was also performed with the data coming from microarray experiments monitoring the amounts of mRNA [15]. Although the two datasets were made differently in different times and different experimental setups, and their time series are not correlated, the results of the simulation produce qualitatively the same results. Both show that the influence of the phosphatases are crucial but can influence the amount of free SigB in only a certain range, and comparable results can also be observed for the other reactions. This provides quite an important conclusion, namely, that for the purposes of modeling, most important are the mutual relations among the components of the system within one dataset, rather than whether the experiment was conducted on the transcriptome or proteome level. The models conducted on a transcriptome level are often criticized for inconsistency with proteins that are actually the molecules that interact. The results produce the same conclusions for both types of experiments.

## 5. Conclusions

In summary, the analysis of the SigB regulatory network presented here provides insights into the quantitative principles of regulation of a group of sigma factors to which SigB belongs. It provides a methodology for the discovery of quantitative principles for factors whose qualitative aspects have already been established.

## Figures and Tables

**Figure 1 biology-13-00614-f001:**
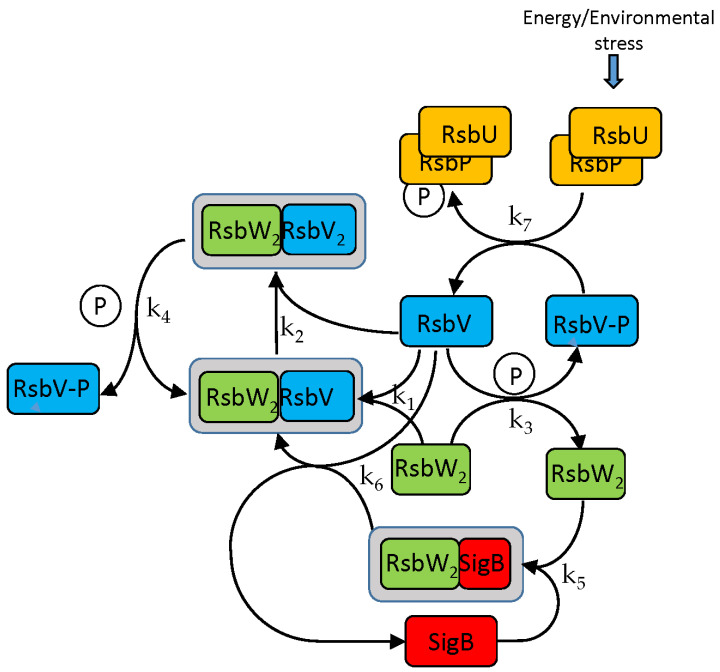
SigB regulatory network. Anti-anti-sigma factor RsbV is dephosphorylated by two phosphatases, RsbP and RsbU. Dephosphorylated RsbV binds to the RsbW_2_ anti-sigma factor dimer to release SigB bound in a complex with RsbW_2_. RsbW also has a protein kinase activity that phosphorylates and inactivates RsbV. Complete description in the form of chemical Equations (ch1)–(ch8) is shown below. k_1_–k_7_ are the corresponding kinetic constants. Grey frame represents complexes.

**Figure 2 biology-13-00614-f002:**
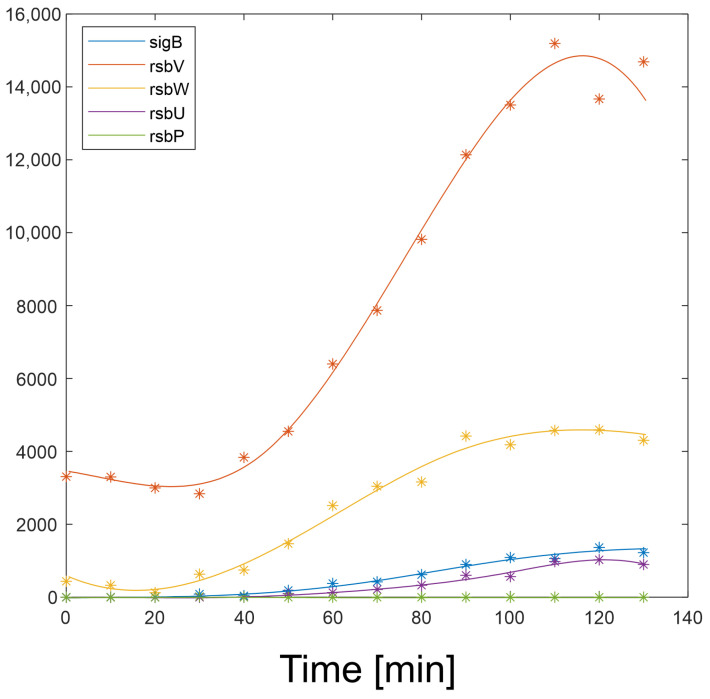
Measured (*) and approximated (solid line) expression profiles of genes of the SigB regulatory network. The original values were divided by 3.7 × 10^4^ in order to be able to use the constants published in [13]. It can be seen that the amount of RsbP is close to zero and does not contribute to the control of the network. Vertical axis—absolute intensity.

**Figure 3 biology-13-00614-f003:**
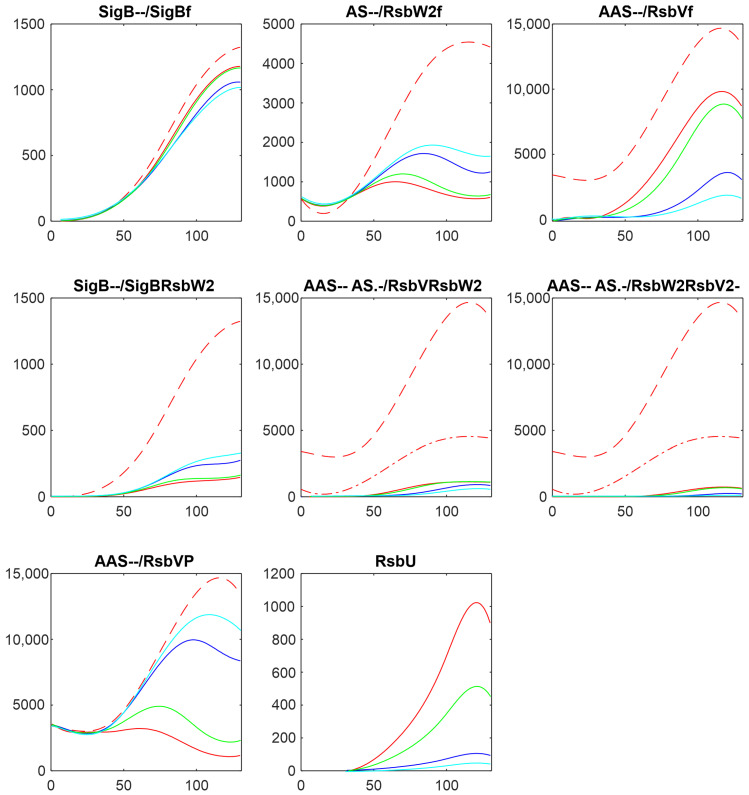
Results of the simulation of changes in the amounts of RsbU on the components of the network. Red—original value, green—RsbU/2, blue—RsbU/10, cyan—RsbU/20. The horizontal axis is given in minutes, vertical axis—absolute intensity. Symbols in the panel captions after the protein name refer to line type: - - dashed; -. dashed–dotted; nothing, full line.

**Figure 4 biology-13-00614-f004:**
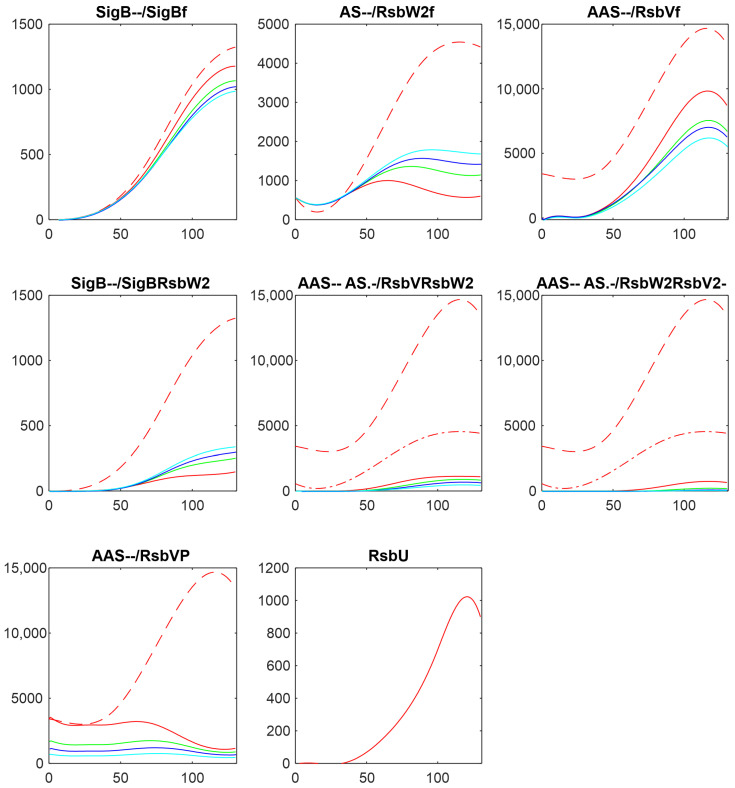
Influence of the decrease in anti-anti-sigma factor RsbV on the amounts of components of the SigB regulatory network. Red—original value, green—AAS/2, blue—AAS/3, cyan—AAS/5. The horizontal axis is given in minutes, vertical axis—absolute intensity. Symbols in the panel captions after the protein name refer to line type: - - dashed; -. dashed–dotted; nothing, full line.

**Figure 5 biology-13-00614-f005:**
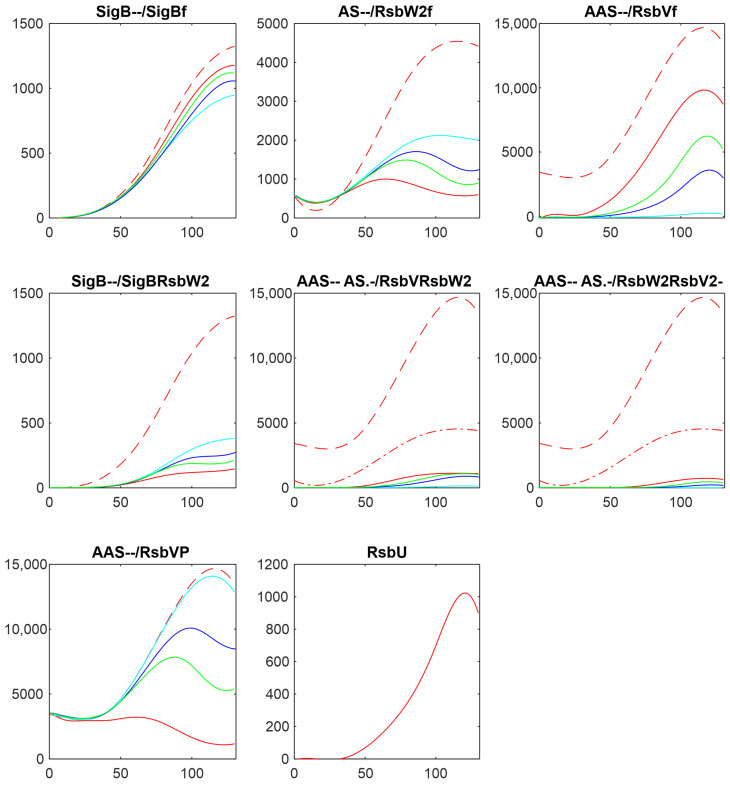
Influence of the rate constant of dephosphorylation of anti-anti-sigma factor RsbV by changing the value of the rate constant of chemical Equation (ch7). Red—original value, green—k_7+_/5, blue—k_7+_/10, cyan—k_7+_/100. The horizontal axis is given in minutes, vertical axis—absolute intensity. Symbols in the panel captions after the protein name refer to line type: - - dashed; -. dashed–dotted; nothing, full line.

**Figure 6 biology-13-00614-f006:**
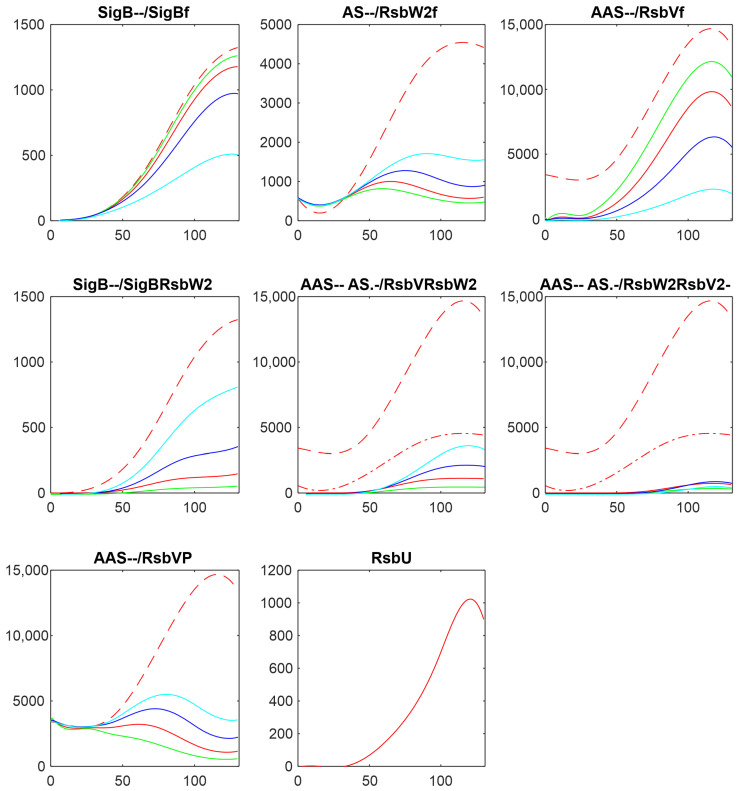
Influence of the changes in the anti-sigma factor RsbW_2_ amounts on the components of the system. Red—original values, green—AS/2, blue—AS*2, cyan—AS*5. The horizontal axis is given in minutes, vertical axis—absolute intensity. Symbols in the panel captions after the protein name refer to line type: - - dashed; -. dashed–dotted; nothing, full line.

**Figure 7 biology-13-00614-f007:**
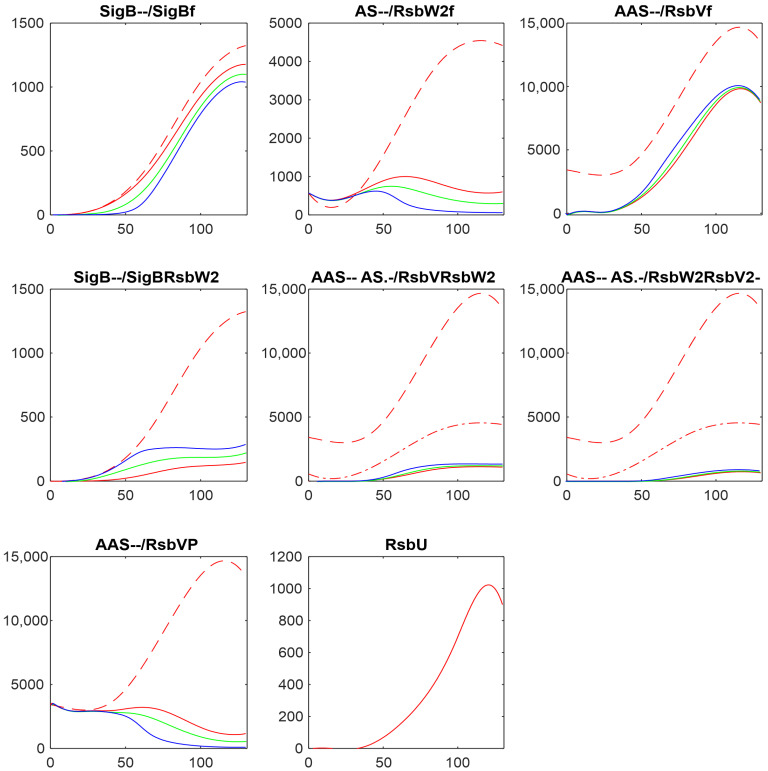
Influence of the shift of reaction of SigB with RsbW_2_ towards formation of the complex SigBRsbW_2_. Red—original values, green—k_5+_*10, k_5−_/10, blue—k_5+_*100, k_5−_/100. The horizontal axis is given in minutes, vertical axis—absolute intensity. Symbols in the panel captions after the protein name refer to line type: - - dashed; -. dashed–dotted; nothing, full line.

**Figure 8 biology-13-00614-f008:**
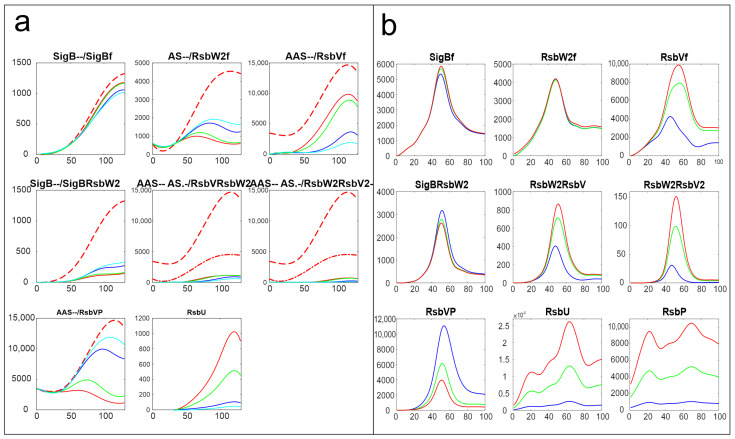
Comparison of the results of simulation of the influence of phosphatases made on proteomic (**a**) and transcriptomic (**b**) data. Panel (**a**)—a copy of Figure 3. Red—original value, green—RsbU/2, blue—RsbU/10, cyan—RsbU/20. Panel (**b**)—a copy of Figure 4 from the publication [15]. Blue—original value, green—RsbU, RsbP*5, red—RsbU, RsbP*10. The horizontal axis is given in minutes, vertical axis—absolute intensity. Symbols in the panel captions after the protein name refer to line type: - - dashed; -. dashed–dotted; nothing, full line.

**Table 1 biology-13-00614-t001:** Kinetic constants [13] converted to min^−1^ to fit the measured expression profiles (y).

Rate Constant	Value
k_1+_	6 × 10^−5^ y^−1^ min^−1^
k_1−_	0.3 min^−1^
k_2+_	6 × 10^−5^ y^−1^ min^−1^
k_2−_	0.3 min^−1^
k_3+_	0.6 min^−1^
k_4+_	0.6 min^−1^
k_5+_	6 × 10^−5^ y^−1^ min^−1^
k_5−_	0.3 min^−1^
k_6+_	3 × 10^−4^ y^−1^ min^−1^
k_6−_	3 × 10^−4^ y^−1^ min^−1^
k_7+_	3 y^−1^ min^−1^

## Data Availability

*B. subtilis* proteomic data were downloaded from PRIDE accession number PXD048690.

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
