# Peer review of "Quantitative Aspect of Bacillus subtilis σB Regulatory Network on a Proteome Level—A Computational Simulation"

_biology, 2024, doi:10.3390/biology13080614_

Round 1

Reviewer 1 Report

Comments and Suggestions for Authors

The manuscript is of interest because relay on the importance of the SigB, the first alternative sigma factor described in bacteria. In B. subtilis SigB controls more than 150 genes playing crucial roles, and activates as say the author the transcription of a diverse set of genes, enabling the bacterium to adapt to a wide range 31 of environmental stresses, including heat, osmotic stress or nutrient deprivation. In this sense the author can explain if the model is sensitive to the general stress response as its mentioned in introduction or one in particular. In abstract, the author mentioned…The solution of these equations allowed the simulation of the kinetic behavior of the network and its components under real conditions…But what is a real condition? This is because in general terms Bacillus subtilis is used from different studies among, basic studies, metabolites production, interaction with others organism including microorganisms, plants, etc. Hence, in each study the real condition can be different.

Author Response

Comment1: The manuscript is of interest because relay on the importance of the SigB, the first alternative sigma factor described in bacteria. In B. subtilis SigB controls more than 150 genes playing crucial roles, and activates as say the author the transcription of a diverse set of genes, enabling the bacterium to adapt to a wide range 31 of environmental stresses, including heat, osmotic stress or nutrient deprivation. In this sense the author can explain if the model is sensitive to the general stress response as its mentioned in introduction or one in particular. In abstract, the author mentioned…The solution of these equations allowed the simulation of the kinetic behavior of the network and its components under real conditions…But what is a real condition? This is because in general terms Bacillus subtilis is used from different studies among, basic studies, metabolites production, interaction with others organism including microorganisms, plants, etc. Hence, in each study the real condition can be different.

Response: The real conditions are given in the time series of gene expression used in the particular simulation.

Action taken: the sentence in the abstract “The solution of these equations allowed the simulation of the kinetic behavior of the network and its components under real conditions“. Was changed to: “The solution of these equations allowed the simulation of the kinetic behavior of the network and its components under real conditions reflected in the time series of protein expression.“

Reviewer 2 Report

Comments and Suggestions for Authors

Comments: This MS is further study for proteome level of sigma factor’s network regulation from previous study for transcriptomic level at 2022 report in same journal (Biology. 11, 1729). Thus, many parts of full MS are repeated, for examples, Figure 1, the model, and so on. Actually, the computational simulation needs to confirm comparing with real practice data. In this case, protein producing work, for example, is mass-spectrometry dataset. Because, the general prokaryotic gene expression conception that is transcription and translation are coupled and especially, Bacillus subtilis has multifaceted growth phases such as spore and vegetable cells. This analytic approache is very interesting model for gene expression controlling among target sigma factor and its anti-sigma factors and anti-anti-sigma factors. The author concluded that this model simulation tool support strongly confirmation to experimental work and despite different experimental setup conditions between transcriptomic (Keijser et al., 2007 with full genes microarray data in time course) and proteomic analysis (mass spectrometry data can’t open from PRIDE), the final results indicated same results. It is really surprised and interested finding. Because, the transcriptomic data obtained germination phase through after heat activation of time zero from spore during first 100 min. However, here both simulational- and experimental cases were not supported in present form and there are no answers about these correlated results at zoom in on sigB-RsbV,W,U, and P. Thus, the MS need to rewrite centered with shorten, and core finding and more convincing data.

Minor comments:

-       PRIDE data are not accessible

-       Fig.1 and figure legend are completely duplicated from previous paper (Biology. 11, 1729). What is the “P” source in RsbV-P?

-       In Fig. 2-5, need to add Y-axis unit. (for example, related expression revels)

-       In line 24, what is this “experimental data”?

-       In line 64, previous data should be cited.

-       In line 65, what is the “heat shock method”? Need to explain in M&M as like previous work. Did time-course Mass-spectrometry conducted and was deposited in PRIDE? (M&M comment is added in below)

-       Maybe reference #16 need to update for “Splinefit”

-       In Fig. 3, what are the dot and semi-dot curves? Might be dashed line is a theoretical curve?

-       In line 152, change to RsbV for anti-anti-sigma factor.

-       In Fig. 4, there is wrong figure title.

-       In M&M, data acquisition: Supplementary file indicated 5 genes expression levels in time course with 1 min intervals. That is simulated using B-Spline coding tool. In total, author must be explained more detailly how dataset mined and modified from MS-spectrometry dataset. What is real experimental condition? Spore fraction, heat shock method, culture medium, and real samples preparation intervals, and finally how to correlated between simulated data and real protein expression levels.

-       There are many repeated sentences with previous author’s paper in Biology:

Lines 8-10, 24-24 in abstract

Lines 84-88, Fig. 1 and its legend

Lines 92-115

Lines 117

Lines 249-252

-       Need to explain kinetic constants were changed in several factors in Table 1.

Comments on the Quality of English Language

The MS need to rewrite centered with shorten, and core finding and more convincing data. Please check conceptional keypoints of your work.

Author Response

Comment 1: This MS is further study for proteome level of sigma factor’s network regulation from previous study for transcriptomic level at 2022 report in same journal (Biology. 11, 1729). Thus, many parts of full MS are repeated, for examples, Figure 1, the model, and so on. Actually, the computational simulation needs to confirm comparing with real practice data. In this case, protein producing work, for example, is mass-spectrometry dataset. Because, the general prokaryotic gene expression conception that is transcription and translation are coupled and especially, Bacillus subtilis has multifaceted growth phases such as spore and vegetable cells. This analytic approache is very interesting model for gene expression controlling among target sigma factor and its anti-sigma factors and anti-anti-sigma factors. The author concluded that this model simulation tool support strongly confirmation to experimental work and despite different experimental setup conditions between transcriptomic (Keijser et al., 2007 with full genes microarray data in time course) and proteomic analysis (mass spectrometry data can’t open from PRIDE), the final results indicated same results. It is really surprised and interested finding. Because, the transcriptomic data obtained germination phase through after heat activation of time zero from spore during first 100 min. However, here both simulational- and experimental cases were not supported in present form and there are no answers about these correlated results at zoom in on sigB-RsbV,W,U, and P. Thus, the MS need to rewrite centered with shorten, and core finding and more convincing data.

Response: The repeated parts of the manuscript are necessary to present the model. As the model is the same as in the previous version where the data came from the transcriptomic experiment, the definition and figures have to be the same. It can also be done so that the model definition and figures are only referenced, but in my opinion the paper will be difficult to understand. Nobody would like to heve two papers open at once to read the later. For this reason I decided to include this part also to the current manuscript.

My apologies, but I don’t understand the comment “Because, the transcriptomic data obtained germination phase through after heat activation of time zero from spore during first 100 min. However, here both simulational- and experimental cases were not supported in present form and there are no answers about these correlated results at zoom in on sigB-RsbV,W,U, and P.”. The functionality of the network controlling SigB is given by the relations among the members of the network. Their amount is given by the conditions in which the experiment was conducted. Therefore the conditions are actually not important as they are already reflected in the time series.

Minor comments:

Comment 2: PRIDE data are not accessible

Response: PRIDE data might not be accessible due to delayed publication of the original paper analysing the dataset. Now the data are fully accessible and reference to the original paper has been added.

Action taken: Reference to the paper analysing the original data was added at line 65 “ The basic analysis of the dataset is given in Pospisil et al. [16].”

(We are happy to inform you that your ProteomeXchange dataset has been successfully made public via the PRIDE database.

Below are details:

 ProteomeXchange title: Whole proteome analysis of germinating Bacillus subtilis 168

 ProteomeXchange accession: PXD048690

 PubMed ID: Not applicable

 Publication DOI: 10.1002/PMIC.202400031

 Project Webpage: http://www.ebi.ac.uk/pride/archive/projects/PXD048690

 FTP Download: https://ftp.pride.ebi.ac.uk/pride/data/archive/2024/07/PXD048690

Thank you for choosing PRIDE and ProteomeXchange for the dissemination of your data, and looking forward to receiving more of your data in the future.)

Comment 3:-       Fig.1 and figure legend are completely duplicated from previous paper (Biology. 11, 1729). What is the “P” source in RsbV-P?

Response: Please see explanation above. P – is a shortcut to phosphate which is considered to be available in access.

Comment 4:-       In Fig. 2-5, need to add Y-axis unit. (for example, related expression revels)

Action taken: Units were added to the figure legends.

Comment 5:-       In line 24, what is this “experimental data”?

Response: line 24 is “whether the data come from the mRNA or protein level. In summary, the computational results”. I don’t see the mentioned term.

Comment 6:-       In line 64, previous data should be cited.

Action taken: reference was added.

Comment 7:-       In line 65, what is the “heat shock method”? Need to explain in M&M as like previous work. Did time-course Mass-spectrometry conducted and was deposited in PRIDE? (M&M comment is added in below)

Action taken: Reference to the original paper where the experiment is described in detail was added.

Comment 8:-       Maybe reference #16 need to update for “Splinefit”

Action taken: correction was made

-       In Fig. 3, what are the dot and semi-dot curves? Might be dashed line is a theoretical curve?

Response: The meaning is given in the caption of each figure. Eg. AAS - - AS .-

Action taken: a sentence “Symbols in the panel captions after protein name refer to line type: . - - dashed, -. dashed dotted, nothing, full line.” Was added to legends of the figures.

Comment 9:-       In line 152, change to RsbV for anti-anti-sigma factor.

Response: there is probably a mismatch in line numbering. Line 152 is: “Figure 4. Influence of decrease anti-anti- sigma factor RsbU on the amounts of components of the”. RsbV is not mentioned.

Comment 10:-       In Fig. 4, there is wrong figure title.

Response: Thanks.

Action taken: RsbU in the figure title was changed to RsbV

Comment 11:-       In M&M, data acquisition: Supplementary file indicated 5 genes expression levels in time course with 1 min intervals. That is simulated using B-Spline coding tool. In total, author must be explained more detailly how dataset mined and modified from MS-spectrometry dataset. What is real experimental condition? Spore fraction, heat shock method, culture medium, and real samples preparation intervals, and finally how to correlated between simulated data and real protein expression levels.

Response: The data in the supplement were included in order to make possible to repeat the simulation. The requested details are given in the paper whose reference was added, and where all the experimental details can be found.

Comment 12:-       There are many repeated sentences with previous author’s paper in Biology:

Lines 8-10, 24-24 in abstract

Lines 84-88, Fig. 1 and its legend

Lines 92-115

Lines 117

Lines 249-252

Response: As this paper is a reconstruction of the model for different type of data some overlaps between these two papers are natural. Changing the text just in order to make the two texts different is in my opinion superfluous.

Comment 13:-       Need to explain kinetic constants were changed in several factors in Table 1.

Response: The constants were adapted to the scale of the given dataset. The constants are varied during the simulation anyway.

Reviewer 3 Report

Comments and Suggestions for Authors

Abstract

Line 22 -> "datasets)" -> datasets -> Correct or review brackets;

Introduction

Line 34 -> "et al." -> Shall be typed in italics, review whole manuscript;

Line 38 -> "RsbW ." -> RsbW. -> Correct spacing;

Line 41 -> Correct extra space before SigB-dependent;

Line 53 -> Add full stop after "[15]" and begin new sentence;

Line 55 -> Accession number PXD048690 search at PRIDE Archive gives no result. Could you please provide more detail how to find these data at PRIDE, such as webpage adress and date of acession?; 

Materials and Methods

Line 65 -> "Briefly - The" -> Briefly, the -> Correct;

Line 66 -> ", B. subtilis culture grew in defined liquid medium." -> Review this sentence because it is disconnected;

Line 66 -> ", B. subtilis culture grew in defined liquid medium." -> This seems to be with a lower typing size? Why? -> Review and correct;

Line 69 -> "were identified quality control passed 2,063 unique proteins" -> Review english grammar for this sentence, which is lacking sense;

Line 71 -> "subsmapled" -> What is this? Subsampled? -> Review or correct;

Results and Discussion

Line 11 -> "k7are" -> k7 are -> Add spacing, correct;

Line 117 -> What is plotted at Figures 2-8 X axis? Is it an arbitrary unit, relative expression, fold increase? Please indicate it in figures 2-8 legends; 

Line 132 -> "RsbUare" -> RsbU are -> Add spacing, correct;

Line 134, Figures 2-8 -> What exactly each graph is showing? Please review graph naming written above each graph, because it is not clear. For instance, "AAS- -/RsbVP", means a ratio of AAS and RsbVP? Please explain it in legends for all figures; You may add letters to each graph, mentioning them in text so readers can more easily follow the explanation; Other possibility is to show only (or highlight) graphs directly affected by the simulation performed, and, if not showing all, present the other ones in supplementary file; In addition, explain in all legends what are dashed red lines shown in graphs;

Line 137 -> "Figure 2" -> Figure 3, right? -> Review and correct;

Line 139 -> Review spacing at this text line;

Line 139 -> "freed" -> released -> review;

Line 150 -> "2,3,and" -> review and correct spacing;

Line 152 -> "RsBU" -> RsbV, right? -> Review and correct;

Line 166 -> Cyan naming for k7/100 condition is lacking -> Correct;

Line 175-176 -> "as the main player" -> it is typed twice -> Review and correct;

Line 186 -> "The effect was large." -> Please review for 'The simulated effect was large' OR 'The simulation resulted a large effect', as it should be noted to readers that what is being done/shown are simulations;

Line 186-187 -> Please connect or condense last two sentences;

Line 187 -> "completely" -> How can you state this? As result depicted in Figure 6, SigBf is at 500 (cyan line), right? Could you better explain it in text, please?;

Line 205 -> "here RsbU" -> RsbU, here, and RsbU and RsbP, for the...-> Correct or review grammar concordance in whole sentence;

Lines 207-209 -> Review english for whole sentence describing Figure 8, in order to be clearer for readers;

Line 210 -> "on the final amounts on the amounts" -> Please review and correct;

Line 212 -> "It shows that the important are the..." -> It shows that the 'most important/critical aspect/s is/are' the... -> Review grammar english and correct, because the idea is not clear here;

Conclusions

Line 221 -> "system what is a trivial" -> system, which is a trivial -> Review and correct;

Line 226 -> Please add RsbW after anti-sigma factor;

Line 227 -> Please review/explain the term 'completely" as commented above;

Line 242 -> ", comparable" -> , and comparable;

Lines 221-248 -> This excerpt is discussion, please review; Conclusions are stated from 249 to 252;

Comments on the Quality of English Language

Comments are pointed above. 

Author Response

Comment1: Line 22 -> "datasets)" -> datasets -> Correct or review brackets;

Action taken: Thanks, corrected.

Comment 2: Line 34 -> "et al." -> Shall be typed in italics, review whole manuscript;

Action taken: corrected in the whole manuscript.

Comment 3: Line 38 -> "RsbW ." -> RsbW. -> Correct spacing;

Action taken: corrected.

Comment 4: Line 41 -> Correct extra space before SigB-dependent;

Action taken: corrected in whole manuscript

Comment 5: Line 53 -> Add full stop after "[15]" and begin new sentence;

Action taken: corrected.

Comment 6: Line 55 -> Accession number PXD048690 search at PRIDE Archive gives no result. Could you please provide more detail how to find these data at PRIDE, such as webpage adress and date of acession?; 

Action taken: Now the dataset under the reference number is accessible.

Comment 7: Line 65 -> "Briefly - The" -> Briefly, the -> Correct;

Action taken: corrected.

Comment 8: Line 66 -> ", B. subtilis culture grew in defined liquid medium." -> Review this sentence because it is disconnected;

Action taken: ?Full stop added after “…. The heat shock method”.

Comment 9: Line 66 -> ", B. subtilis culture grew in defined liquid medium." -> This seems to be with a lower typing size? Why? -> Review and correct;

Action taken: corrected.

Comment 10: Line 69 -> "were identified quality control passed 2,063 unique proteins" -> Review english grammar for this sentence, which is lacking sense;

Action taken: comma added. Sentence changed to: “A total of 2,191 proteins were identified, quality control passed 2,063 unique proteins.”

Comment 11: Line 71 -> "subsmapled" -> What is this? Subsampled? -> Review or correct;

Response: a term used for sampling intervals smaller than in the original dataset.

Comment 12: Line 11 -> "k7are" -> k7 are -> Add spacing, correct;

Action taken: corrected.

Comment 13: Line 117 -> What is plotted at Figures 2-8 X axis? Is it an arbitrary unit, relative expression, fold increase? Please indicate it in figures 2-8 legends; 

Action taken: vertical axis description was added.

Comment 14: Line 132 -> "RsbUare" -> RsbU are -> Add spacing, correct;

Action taken: corrected.

Comment 15: Line 134, Figures 2-8 -> What exactly each graph is showing? Please review graph naming written above each graph, because it is not clear. For instance, "AAS- -/RsbVP", means a ratio of AAS and RsbVP? Please explain it in legends for all figures; You may add letters to each graph, mentioning them in text so readers can more easily follow the explanation; Other possibility is to show only (or highlight) graphs directly affected by the simulation performed, and, if not showing all, present the other ones in supplementary file; In addition, explain in all legends what are dashed red lines shown in graphs;

Response: the symbols after protein/symbol name refer to line type. - - dashed, -. Dashed dotted, nothing, full line.

Action taken: explanation was added to figure legends.

Comment 16: Line 137 -> "Figure 2" -> Figure 3, right? -> Review and correct;

Action taken: corrected.

Comment 17: Line 139 -> Review spacing at this text line;

Action taken: corrected.

Comment 18: Line 139 -> "freed" -> released -> review;

Action taken: corrected.

Comment 19: Line 150 -> "2,3,and" -> review and correct spacing;

Action taken: corrected.

Comment 20: Line 152 -> "RsBU" -> RsbV, right? -> Review and correct;

Action taken: corrected.

Comment 21: Line 166 -> Cyan naming for k7/100 condition is lacking -> Correct;

Action taken: corrected.

Comment 22: Line 175-176 -> "as the main player" -> it is typed twice -> Review and correct;

Action taken: corrected.

Comment 23: Line 186 -> "The effect was large." -> Please review for 'The simulated effect was large' OR 'The simulation resulted a large effect', as it should be noted to readers that what is being done/shown are simulations;

Action taken: The sentence was changed to: “'The simulation resulted a large effect”.

Comment 24: Line 186-187 -> Please connect or condense last two sentences;

Action taken: The sentence was changed to: “'The simulation resulted a large effect, increasing amount of ant-sigma factor can attenuate free SigB completely.”

Comment 25: Line 187 -> "completely" -> How can you state this? As result depicted in Figure 6, SigBf is at 500 (cyan line), right? Could you better explain it in text, please?;

Action taken: The sentence was changed to: “, increasing amount of ant-sigma factor can result in almost complete attenuation of SigB.”

Comment 26: Line 205 -> "here RsbU" -> RsbU, here, and RsbU and RsbP, for the...-> Correct or review grammar concordance in whole sentence;

Action taken: changed to: “RsbU, here,  and RsbU and RsbP, for the transcriptomic experiment”.

Comment 27: Lines 207-209 -> Review english for whole sentence describing Figure 8, in order to be clearer for readers;

Action taken: the text was changed to: “Figure 8 shows similar trends - the increasing/decreasing amounts of phosphatases are reflected in the same way in both experiments. The line color order in the graphs of both panels exhibit the same trends.”

Comment 28: Line 210 -> "on the final amounts on the amounts" -> Please review and correct;

Action taken: corrected.

Comment 29: Line 212 -> "It shows that the important are the..." -> It shows that the 'most important/critical aspect/s is/are' the... -> Review grammar english and correct, because the idea is not clear here;

Action taken: The sentence was changed to: “It shows that the crucial aspect of the network is the mutual relationship among the molecules and their expression time series within the system, rather than whether the data comes from proteomic or transcriptomic sources.”

Comment 30: Line 221 -> "system what is a trivial" -> system, which is a trivial -> Review and correct;

Action taken: corrected.

Comment 31: Line 226 -> Please add RsbW after anti-sigma factor;

Action taken: corrected.

Comment 32: Line 227 -> Please review/explain the term 'completely" as commented above;

Action taken: The sentence was changed to: “If one wants to control SigB regulation, the most effective way is to overproduce the anti-sigma factor, which can ultimately stop the production of SigB entirely.”

Comment 33: Line 242 -> ", comparable" -> , and comparable;

Action taken: corrected.

Lines 221-248 -> This excerpt is discussion, please review; Conclusions are stated from 249 to 252;

Action taken: corrected.

Round 2

Reviewer 2 Report

Comments and Suggestions for Authors

The revised MS was improved some paragraphs and figure legends but not shortened in Fig 1 and all equations which were repeated in previous report. I know that sometimes the background data sets are key elements for understanding in full view of presentation. Nevertheless, my opinion is that most important things are fully explanations between previous work and present work, especially here is transcriptomics and proteomics data handling including kinetic constant conversion and differential factors in gene expression contain protein amounts and activity. Successful computational modeling expended scientific insights in limited experimental analysis. Thus, this work is a challenge to define understanding of regulatory network in well concepted SigB action in Bacillus subtilis.

Minor comments:

1.     Line 83, delete Discussion title

2.     Line 118, ki change to k1

3.     Lines 135-136, should be improved this sentence. How to explain activity≠amount

4.     Lines 250-251, need to improved the conceptional sentence. Anti-sigma factor stops the production of SigB entirely? Its means that anti-sigma factor SigW2 directly repressed the expression of sigB. Is it conceptionally correct?

5.     Simple summary should be combind to Conclusion section.

Comments on the Quality of English Language

MS is well written.

Author Response

Comment 1.     Line 83, delete Discussion title

Action taken: deleted

Comment 2.     Line 118, ki change to k1

Action taken: ki changed to k1

Comment 3.     Lines 135-136, should be improved this sentence. How to explain activity≠amount

Action taken: sentence changed to “,The free SigB amount is determined by the phosphatases RsbU and RsbP, which dephosphorylate the anti-anti-sigma factor RsbV. RsbV then binds to the anti-sigma factor, freeing SigB.”

Comment 4.     Lines 250-251, need to improved the conceptional sentence. Anti-sigma factor stops the production of SigB entirely? Its means that anti-sigma factor SigW2 directly repressed the expression of sigB. Is it conceptionally correct?

Action taken: thew sentence was changed to: “If one wants to control SigB regulation, the most effective way is to overproduce the anti-sigma factor, which can attenuate the production of SigB entirely

Comment 5.     Simple summary should be combind to Conclusion section.

Responese: Position and content of the simple summary is dictated by the publisher.

Reviewer 3 Report

Comments and Suggestions for Authors

Line 72 -> Delete hyphen and add comma after 'Briefly';

Line 79 -> It remains written 'subsmapled' -> subsampled, right? -> I see that you are meaning sampling intervals smaller than in the original dataset, but that word is wrongly typed, quite surely, please review and correct;

Line 144 -> Extra dot before "Symbols', please correct;

Line 144 -> 'Symbols in the panel captions after protein name refer to line type: . - - dashed, -. dashed-dotted, nothing, full line.' -> this sentence does not improve legends clarity and suggestion is to remove it from all figure legends;

Line 202 -> 'The simulation... -> The simulation -> delete quotation mark before The;

Line 229 -> 'The same trends can be seen for the other simulations as well (data not shown)' -> Should present these results, even as a supplementary file, or the suggestion is to omit this sentence.

Line 251-252 -> Check for extra final dots at the end of the sentence;

Author Response

Comment 1: Line 72 -> Delete hyphen and add comma after 'Briefly';

Action taken: changes made.

Comment 2: Line 79 -> It remains written 'subsmapled' -> subsampled, right? -> I see that you are meaning sampling intervals smaller than in the original dataset, but that word is wrongly typed, quite surely, please review and correct;

Action taken: yes, corrected.

Comment 3: Line 144 -> Extra dot before "Symbols', please correct;

Action taken: change made.

Comment 4: Line 202 -> 'The simulation... -> The simulation -> delete quotation mark before The;

Action taken: change made.

Comment 5: Line 229 -> 'The same trends can be seen for the other simulations as well (data not shown)' -> Should present these results, even as a supplementary file, or the suggestion is to omit this sentence.

Action taken:  sentence omitted.

Comment 6: Line 251-252 -> Check for extra final dots at the end of the sentence;

Action taken: dot deleted.